# Antioxidant Biomolecules and Their Potential for the Treatment of Difficult-to-Treat Depression and Conventional Treatment-Resistant Depression

**DOI:** 10.3390/antiox11030540

**Published:** 2022-03-11

**Authors:** María Eugenia Riveros, Alba Ávila, Koen Schruers, Fernando Ezquer

**Affiliations:** 1Centro de Fisiología Celular e Integrativa, Facultad de Medicina Clínica Alemana-Universidad del Desarrollo, Santiago 7710162, Chile; 2Centro de Medicina Regenerativa, Facultad de Medicina Clínica Alemana-Universidad del Desarrollo, Santiago 7710162, Chile; aavilas@udd.cl; 3Department of Psychiatry and Neuropsychology, Maastricht University Medical Center, 6229 Maastricht, The Netherlands; koen.schruers@maastrichtuniversity.nl

**Keywords:** oxidative stress, neuroinflammation, major depressive disorder, mesenchymal stem cells, exosomes, plant extracts

## Abstract

Major depression is a devastating disease affecting an increasing number of people from a young age worldwide, a situation that is expected to be worsened by the COVID-19 pandemic. New approaches for the treatment of this disease are urgently needed since available treatments are not effective for all patients, take a long time to produce an effect, and are not well-tolerated in many cases; moreover, they are not safe for all patients. There is solid evidence showing that the antioxidant capacity is lower and the oxidative damage is higher in the brains of depressed patients as compared with healthy controls. Mitochondrial disfunction is associated with depression and other neuropsychiatric disorders, and this dysfunction can be an important source of oxidative damage. Additionally, neuroinflammation that is commonly present in the brain of depressive patients highly contributes to the generation of reactive oxygen species (ROS). There is evidence showing that pro-inflammatory diets can increase depression risk; on the contrary, an anti-inflammatory diet such as the Mediterranean diet can decrease it. Therefore, it is interesting to evaluate the possible role of plant-derived antioxidants in depression treatment and prevention as well as other biomolecules with high antioxidant and anti-inflammatory potential such as the molecules paracrinely secreted by mesenchymal stem cells. In this review, we evaluated the preclinical and clinical evidence showing the potential effects of different antioxidant and anti-inflammatory biomolecules as antidepressants, with a focus on difficult-to-treat depression and conventional treatment-resistant depression.

## 1. Introduction

From 1990 to 2017, cases of major depressive disorder (MDD) increased worldwide by nearly 50% [1]. Depression is among the three projected leading causes of burden of disease for 2030 [2], together with ischaemic heart disease, a pathology that in itself also has an increased risk in depressive patients. Not included in the previously mentioned projection is the devastating effect on mental health that the COVID-19 pandemic will probably leave behind: among a sample of COVID-19 survivors, 55% showed psychiatric sequelae, including MDD in 30% of the cases [3].

Pharmacological approaches are the most usual treatments for MDD, with selective serotonin reuptake inhibitors (SSRI) being used in most cases. However, around 50% of MDD patients do not respond to this first-line treatments and require a second-line treatment to achieve remission [4]. In addition, antidepressant drugs commonly used are associated with several adverse effects that have led to their early discontinuation [5,6]. Furthermore, it is urgent to improve the present capacity for diagnosis and treatment of this disease since no reliable biomarkers are available, nor is there a guide to predict the response to treatment and the course of the disease [7]. In this sense, a critical limitation is that regardless of the increasing research efforts, the pathophysiology of depression is still not completely understood. Nevertheless, there are active lines of research devoted to deciphering the cellular and molecular mechanisms involved in the development of depression. In particular, in the last years, there has been an increase in research directed toward the unveiling of the role of oxidative stress and neuroinflammation in depression, and how those interact to decrease monoamine neurotransmission and neurotrophic factor levels and induce mitochondrial dysfunction, alterations known to be involved in the onset of MDD [8]. 

A key factor connecting all the previous variables is stress and the concomitant dysregulation of the hypothalamus–pituitary axis stress response system [9,10]. In this review, we will discuss how chronic unpredictable stress is related to the alterations associated with the depressive state.

The role of diet in depression risk has been deeply explored, showing that anti-inflammatory and antioxidant diets may prevent and/or help in the treatment of depressive disorders, which has led to the exploration of antioxidant and anti-inflammatory molecules derived from plants for the treatment of MDD.

In this review, we will first explore the spectrum of available treatments for depression and their potential action on oxidative stress in the brain of depressed patients. Then, we will briefly discuss oxidative damage in relation to depressive disorder and how oxidative stress is associated with the inflammatory sate commonly seen in the depressive condition. Finally, we will review: (i) some plant molecules with antioxidant action that are proposed as potential antidepressants, and (ii) molecules produced by the anti-inflammatory and antioxidant mesenchymal stem cells that could act as new therapeutic options for the treatment of this devastating disease.

## 2. Current Treatments for MDD and Treatment Challenges

In most of cases, treatment for MDD consists of psychological or pharmacological treatments, or a combination of both interventions. A large majority of patients prefer psychological treatment over drug-based treatment [11]. Nevertheless, patients with more severe forms of depression have been reported to prefer pharmacological treatment [12].

There are different antidepressant agents with different mechanisms of action, but they appear to share some common targets. For example, there is substantial preclinical and clinical evidence showing that different antidepressant drugs, including the inhibitors of monoamine oxidase and selective inhibitors of serotonin reuptake, have an antioxidant action, apparently mediated by their ability to increase the organism antioxidant capacity [13]. Using these first-line antidepressant drugs, the chances of patients getting into remission in the first attempt is about 36%. After an unsuccessful attempt in treatment due to the lack of response or intolerable adverse effects, further steps can be taken by changing the drug used or augmenting the dose of the drugs. With these further steps, the chances of remission are increased, leading to an overall remission rate of approximately 67% [14]. Nevertheless, 30% of the patients do not achieve remission with the available antidepressant drugs, and a big percentage of treated patients present adverse reactions, mainly associated with the binding of the drugs to serotoninergic, noradrenergic, adrenergic, cholinergic, or histaminergic receptors [15], thus greatly limiting adherence to the treatments and increasing the social burden of the disease [16].

In this regard, investigating whether the antioxidant effects of current antidepressant treatments is sufficient to treat depression symptoms and searching for new antidepressant drugs directed toward the reduction of oxidative stress may help to avoid adverse outcomes and increase adherence [16].

On the other hand, there are many established psychological treatments for MDD. Some of them have been more extensively used and studied; these include cognitive behaviour therapy, behavioural activation therapy, interpersonal psychotherapy, problem-solving therapy, and non-directive counselling, all of which have comparable efficacy in the treatment of depression [17]. Psychological interventions for MDD have a response rate of nearly 50% [18]. These interventions are believed to work by increasing emotional regulation, in consistency with the reported effect of psychotherapy in the reduction of the activation of specific brain structures such as the anterior cingulate cortex, inferior frontal gyrus, and the insula [19]. It is noteworthy that psychotherapy as well as antidepressant drugs reduce peripheral oxidative stress in MDD patients [20].

Trials comparing psychotherapy treatments with antidepressant drugs show either no difference between them or a moderate advantage of psychotherapy over pharmacological treatments [21]. Additionally, second-generation antidepressants have comparatively more adverse effects than psychotherapy, and it is more likely that patients discontinue their treatment because of these adverse effects.

Other evidence-based treatments that can be considered as alternatives to conventional depression treatments are: exercise, acupuncture, yoga, meditation, and selected herbal or omega 3 fatty acid diet supplementation.

Physical exercise by itself is effective for the reduction of depression symptoms, but it is not superior to psychological treatments or antidepressant medication [22]. Exercising induces acute increases in Brain-derived neurotrophic factor (BDNF) levels immediately after its performance, both in normal subjects and in depressed patients [23]. Depressed individuals have lower basal levels of BDNF, and several effective depression treatments are able to increase BDNF levels. However, the latter is only true for BDNF levels measured immediately after exercise but not for basal levels in individuals that exercise frequently.

Yoga has also been shown to be effective in the treatment of depressive symptoms [24]. Naveen et al. reported that three months of yoga practice reduces depressive symptoms and cortisol levels while increasing BDNF basal levels [25]. Furthermore, twelve weeks of yoga and meditation intervention diminished oxidative stress in healthy individuals [26] and in patients with major depression, while increasing BDNF [27]. Likewise, yoga practiced for eight weeks reduced inflammatory markers and depressive symptoms in patients with rheumatoid arthritis and comorbid depression [28]. There is need for further work to determine the effectiveness of yoga as a treatment for depression in the long term. Thus, more randomized control trials with a longer duration as well as more patients are needed. However, in the short term, there is moderate evidence supporting that yoga interventions as an ancillary treatment could be superior compared to usual pharmacological treatments [29].

Manual and electric acupuncture in combination with second-generation antidepressants are more effective in improving depression symptoms to a greater extent than antidepressant treatment alone [30,31]. Electric acupuncture increases glutathione levels (a potent antioxidant molecule) in the urine of depressed patients receiving antidepressant drugs as compared to patients receiving only antidepressant drugs. Acupuncture treatment also affects tryptophan metabolism and fatty acid biosynthesis, which are metabolic changes that could be related with the improvement of sleep and cognitive disturbances as well as with an antioxidant effect [32]. Nevertheless, according to a recent metanalysis, there is still need for more rigorous primary studies to confirm the effectiveness and safety of acupuncture as compared to antidepressants [33]. Acupuncture as opposed to psychological therapy, yoga, or meditation can be tested in animal models, which are valuable tools for studying its possible mechanisms of actions. According to a recent review, acupuncture has been shown in rodent models of depression to significantly reduce the release of a corticotrophin-releasing hormone, which is a marker of chronic stress, as well as to increase the expression of BDNF, to positively affect hippocampal plasticity, and to regulate neurotransmitter levels [34]. Acupuncture can reduce oxidative stress in animal models for different pathologies such as depression in ovariectomized rats where it was shown to be effective in reducing depression-like symptoms [35], but also in other models such as that of multi-infarct rats, a model of dementia [36], and in a poststroke depression model [37].

According to a recent metanalysis of double-blind, randomized, placebo-controlled trials [38], the intake of omega-3 polyunsaturated fatty acids (PUFAs) reduces depressive symptoms. This effect has been observed for eicosapentaenoic acid (EPA) intake; while the intake of pure docosahexaenoic acid (DHA) does not have the same benefits. The amount of omega-3 PUFAs in the membrane of cells competes with omega-6 for the synthesis of anti-inflammatory and pro-inflammatory eicosanoids, respectively; therefore, omega 3 content must be equilibrated with omega 6. Thus, keeping a low omega 6/omega 3 ratio in order to avoid pro-inflammatory conditions [39] and/or omega 3 supplementation can reduce neuroinflammation and oxidative stress [40]. Moreover, telomere length, which is critically associated with inflammation and oxidative stress, is inversely correlated with the omega 6/omega 3 ratio [41].

### Proposed Treatments for Difficult-to-Treat Depression (DTD)

Despite all the advances in treatment alternatives for ameliorating MDD, there are still patients that do not respond to these. Difficult-to-treat depression (DTD) is a term that has been adopted by consensus and is meant to replace the term “treatment-resistant depression”, with the aim of conveying that it is a form of depression that is challenging but not impossible to treat. This term is used for patients with depression symptoms that continue to cause significant burden despite the usual treatment efforts [42].

Different pharmacological approaches for the treatment of DTD such as the augmentation of SRRI or tricyclics dose, or a combination with lithium or atypical antipsychotic drugs have shown promising results that still need further confirmation [43]. Similarly, adding psychotherapy to the antidepressant treatment is beneficial for DTD, but evidence as to whether the switch to psychotherapy is better than maintaining the antidepressant treatment is still lacking [44].

Ketamine is an antagonist of the *N*-methyl-d-aspartate type of glutamate receptor, commonly used as an anaesthetic drug. Nevertheless, it has also shown a fast and robust antidepressant effect when administered in subanaesthetic doses to patients with DTD [45]. Preclinical evidence shows that together with reducing depressive-like symptoms, ketamine reduces oxidative stress and inflammation in the brain [46].

Ketamine is a racemic mixture of esketamine and arketamine. The intranasal administration of esketamine has been approved by the Food and Drug Administration (FDA) for the treatment of DTD; nevertheless, there are concerns regarding its potential for abuse since it also has addictive effects. Preclinical studies in animal models of MDD have pointed out that arketamine could have the advantage of longer lasting and more potent antidepressant effects and a safer profile as compared to esketamine and ketamine [47]. The same potential has also been observed in DTD patients, showing a superior antidepressant effect that could be longer lasting and more potent than the racemic mixture or esketamine alone [48].

Another fast-acting antidepressant drug that has been successfully tested in DTD patients is Psilocybin, a naturally occurring plant alkaloid with psychedelic effects. Psilocybin acts as an agonist for the 2A serotonin receptors [49]. In a clinical study with 20 DTD patients, psilocybin was able to reduce depressive symptoms in one week, and the effects were maintained for three months after the treatment with low doses of psilocybin administered in a supported environment [50]. These promising effects are supported by a recent meta-analysis in which the authors concluded that psilocybin combined with behavioural support may be a safe and effective alternative treatment for depression [51]. Nevertheless, more placebo-controlled trials are needed to validate this treatment, as well as more clinical trials that include DTD patients to validate its effectiveness in this pathology. Interestingly, psilocybin also has a potential as an antioxidant [52], but it has not yet been proven to effectively reduce oxidative stress in the brain of depressed patients.

Electro-convulsive therapy (ECT) has a bad reputation, but when administered with anaesthesia and muscle relaxants using new technologies that deliver ultra-brief pulses, it is a relatively safe and highly effective alternative treatment for severe depression and for patients that do not respond to first-choice treatments such as pharmacotherapy and psychotherapy. The principal drawback of this treatment is that it can produce cognitive side effects such as impairments in autobiographical memory that are potentially long-lasting [53]. Nevertheless, a cognitive function such as processing speed, which is reduced in depressed patients, is increased by ECT [54]. Moreover, attention and verbal memory that are impaired by ECT treatment are usually recovered to baseline levels six months after the treatment [55]. ECT may be used in geriatric depressed patients as a safe alternative when there is not enough response to conventional drugs or when patients cannot tolerate these drugs [56]. It has been reported that in bipolar depressed patients that respond to ECT treatment, the oxidative stress marker malondialdehyde (MDA) is reduced by ECT treatment [57], suggesting that the antioxidative effect of ECT could be relevant for its antidepressant effect.

Neurodegenerative diseases are bidirectionally related with depression [58], and oxidative stress is part of their shared pathophysiology [59]. Patients with comorbid MDD and neurodegenerative diseases such as Parkinson’s, Alzheimer’s, and Huntington’s diseases respond poorly to standard antidepressant treatments and are in higher risk of side effects [58]. Therefore, the search for alternative treatments for depression in patients with neurodegenerative disease is urgent, and oxidative stress appears to be an interesting therapeutic target.

Altogether, the presented evidence shows that many proved treatments for depression and treatment-resistant depression share an antioxidant action. Nevertheless, more preclinical and clinical research is needed to state if this shared antioxidative action is critical to their antidepressant efficacy, and to determine how oxidative status reduction is related to depressive symptom amelioration; additionally, whether this relation is associated with some or all depressive symptoms and different types of depressive disorders also needs to be determined. Moreover, these treatments have not been specifically designed to achieve a reduction in oxidative stress. Thus, directing the treatments towards an emphasis on antioxidation might allow for the reduction of some of the unwanted effects while maintaining or even improving their efficacy.

## 3. Oxidative Damage in the Brain and Its Association with MDD

Oxidative damage is a consequence of the oxygen dependence of cellular metabolism. The presence of oxygen in the internal media is, on the one hand, crucial for survival; on the other hand, it is a menace producing oxidative damage through the generation of free radical species, which have to be counteracted by the presence of potent enzymatic and non-enzymatic antioxidants. These molecules are part of a complex system of structurally diverse functional components comprising endogenous and exogenous antioxidant molecules, as shown in (Figure 1). Therefore, there must be an equilibrium among the production of reactive oxygen species (ROS) and the antioxidant defence response. The brain is especially prone to oxidative stress because it has a high amount of transition metals and polyunsaturated fatty acids that provide a substrate for lipid peroxidation, in addition to its high oxygen consumption rate and limited antioxidant defences [60] (Figure 2).

Mitochondrial function is tightly related to oxidative stress. The electron transport chain in the mitochondria is coupled with the production of ROS-like superoxide radicals and hydrogen peroxide (H_2_O_2_), in addition to its presence in the outer membrane of the mitochondria of enzymes such as monoamine oxidase, which also produce ROS. On the other hand, manganese superoxide dismutase (SOD) and glutathione (GSH) molecules inside the mitochondria act as an antioxidant mechanism [61]. Outside the mitochondria, H_2_O_2_ is enzymatically degraded by catalase, glutathione peroxidase, and peroxiredoxin. Nevertheless, the overproduction of ROS by mitochondria dysfunction or the reduction in antioxidant defence may cause oxidative damage, particularly in the brain where mitochondrial ROS overproduction is involved in many psychiatric and neurodegenerative diseases [62].

It has been consistently reported that the antioxidant capacity, dependent on both enzymatic and non-enzymatic antioxidants, is reduced in depressed patients. For example, levels of glutathione peroxidase [63], vitamin E [64,65], erythrocyte superoxide dismutase, and glutathione reductase [66] are reduced in the blood of depressed patients, but also in their brains [67]. Consistently, the application of chronic unpredictable stress in rats decreases the expression of different antioxidant enzymes in the brain and in the periphery, and these alterations can be reversed by antidepressant treatments [68].

In fact, mitochondrial dysfunction in the brain is associated with depression [69]. Moreover, mitochondria are instrumental in handling the stress response, supplying the increased energy demands during stress and also producing and responding to neuroendocrine and metabolic stress mediators such as glucocorticoids [70]. Thus, brain mitochondria are affected by chronic stress, and alterations in mitochondrial function have been related to stress-induced cognitive and behavioural changes [71].

Genetic vulnerability and stressful life events are important factors that determine depression risk since they interact synergistically [72]. A history of stress increases vulnerability to new stress by exerting epigenetic modification in the risk genes via DNA methylation and miRNA regulation, leading to alterations in the brain, which then increase the vulnerability to developing depressive disorders [73]. Oxidative stress may be involved in the increase in vulnerability, as shown in a rat model of social defeat stress (SDS)—a major acute stress that induces a reduction in BDNF levels, but only in animals vulnerable to depression [74]. This reduction in BDNF levels is maintained for weeks, in association with a greater oxidative stress as compared to animals not vulnerable to depression. In this model, vulnerability is abolished by antioxidant treatments, suggesting that oxidative stress is involved in generating the vulnerability to depression. It is postulated that vulnerable animals have a prolonged oxidative stress response after experiencing acute major stress because the transcription factor that initiates the response to control oxidative stress levels—the nuclear factor erythroid-2 related factor 2 (NrF2)—is downregulated [75]. This downregulation is associated with a reduction in BDNF levels, since BDNF induces NrF2 translocation to the nuclear compartment [75] (Figure 2). Supporting these findings, the induction of NrF2 translocation to the nuclear compartment by stimulating the BDNF receptor TrKB reverses the vulnerability to depression (Bouvier et al., 2017).

It has been reported that chronic mild stress decreases glutathione peroxidase (GSH-Px) activity and glutathione (GSH) and vitamin C levels in the brain [76], which are examples of the enzymatic and non-enzymatic antioxidant mechanisms in the brain, respectively (Figure 2). Furthermore, lipid peroxidation is highly increased in stress-induced depression in different rat tissues, with the brain being the most affected organ [77]. Antidepressant treatment with venlafaxine, an inhibitor of serotonin and norepinephrine reuptake, prevents oxidative stress by potentiating the brain antioxidant defence and reducing stress-induced lipid peroxidation in the brain [76].

Meanwhile, in MDD patients, oxidative stress markers are elevated, and higher baseline levels of F2-isoprostanes, a marker of oxidative stress, is related to a poorer response to antidepressant treatment [78]. On the other hand, patients who respond to the treatments showed reductions in the oxidative markers [78,79].

Therefore, the maintenance of a delicate equilibrium between ROS production and antioxidant defences is essential for correct brain functioning and for the ability of the organism to respond to stress. Hence, oxidative homeostasis alteration is a major player in neuropsychiatric and neurodegenerative diseases, including depression, and represent an interesting target for their treatment.

## 4. Oxidative Stress and Inflammation Crosstalk in the Brain of Depressive Patients

In the last ten years, evidence linking depression to inflammation has been accumulated. It is well-known that inflammatory mediators are elevated in depressed patients; for example, C-reactive protein (CRP) levels are elevated in MDD patients as compared to healthy controls, and a third of them have CRP levels compatible with low-grade inflammation [80]. Similarly, the levels of various interleukins, including tumour necrosis factor alpha (TNF-α) and the soluble intlerleukin-2 receptor (sIL-2R), are elevated in depressed patients [81]. This elevation in cytokine levels is not a generalized response but rather a more specific pro-inflammatory regulation, in which some pro-inflammatory cytokines are elevated and some anti-inflammatory cytokines are reduced in the plasma of depressed patients [82]. Further supporting the idea that inflammation may be involved in generating depression is the data showing that inflammation in elderly patients is associated with depression, but not with Alzheimer disease [83].

The association of certain types of diets with depression may be related to the inflammation induced by gut dysbiosis [84] (Figure 2). Likewise, psychological stress such as marital distress is associated with inflammation in the gut, promoting the translocation of gut bacteria products such as lipopolysaccharide (LPS) to the portal blood, inducing an immune response and a general pro-inflammatory state [85].

Furthermore, higher levels of inflammatory markers are associated with a poorer response to treatment in depressed patients, and treatment success is associated with a reduction in inflammatory markers [86], suggesting that inflammation does not merely coexist with depression or is just a marker for every neuropsychiatric alteration, but rather that it is likely to be related to the manifestation of symptoms of depression. Nevertheless, oxidative stress and neuroinflammation are implicated in many other neuropsychiatric alterations such as Parkinson’s disease or posttraumatic cognitive damage.

In the same sense, experiencing stress and having a history of major depression are associated with metabolic alterations that promote inflammation, showing an intricate, bidirectional relationship between inflammation and depression that is probably mediated by stress [87] (Figure 2).

Treatments with anti-inflammatory molecules may have an anti-depressive potential. This seems to be the case for omega 3 diet supplementation, for example, but when the anti-inflammatory effect is associated with a pro-oxidative action, as is the case for inhibitors of cyclooxygenase 2, there is no anti-depressive potential [88]. Moreover, treatment with pro-inflammatory agents such as interferon alpha, used to treat chronic hepatitis C, frequently induces depression, an outcome more likely in patients with a history of major depression [89]. Therefore, inflammation is not sufficient to induce depression, but it does favour its development.

Preclinical research shows that chronic unpredictable stress used in animal models to induce depression also induces neuroinflammation and oxidative stress. In the same way, inflammation induced by simulating the presence of bacteria with LPS injection can also induce depressive symptoms [90].

Subclinical systemic inflammation is a risk factor for developing depression [91]. Adults that have been exposed to childhood maltreatment are at risk of depression and have increased markers of inflammation [92]. Additionally, childhood maltreatment is associated with mitochondrial malfunctioning and oxidative stress [93].

Oxidative stress can damage mitochondria, and in turn, damaged mitochondria can induce inflammation since the components of the damaged mitochondria are recognized by the immune system [94]. Consequently, mitochondrial damage promotes oxidative stress and is involved in depressive pathology, as previously discussed, generating a vicious cycle that is maintained for a long time.

As mentioned, oxidative stress is increased in depressed patients [95,96], as measured by the increase in different oxidation markers. Examples are the levels in serum and urine of the excretion of F2 isoprostane, a derivate of free radical-mediated lipid peroxidation [97], and the plasmatic levels of the end-product of lipid peroxidation: MDA [64]. The former reflects peripheral oxidation. Interestingly, brain DNA damage induced by oxidative stress is increased in depressed patients as compared to healthy controls, suggesting that oxidative stress-induced damage in oligodendrocytes and consequent white matter alterations might be involved in depression disorder pathogenesis [98].

Interestingly, the reduction in antioxidant defences in depressed patients is strongly associated with elevated cytokine levels in their blood [66]. In fact, ROS can induce the activation of the inflammasomes. These are protein complexes, including Nod-like receptor protein (NLRP) 3 and 6, the NLR family, CARD domain containing 4 (NLRC4), and absent in melanoma 2 (AIM2), that can activate the caspase and interleukin systems, initiating an inflammatory response [99]. Inflammasome activation can initiate a Caspase-1-dependent programmed cell death named pyroptosis, which has been associated with depression [100] (Figure 2). In line with this, it has been reported that melatonin administration can reduce depressive-like behaviour in an animal model of depression by reducing NLRP3 inflammasome activation through the activation of Nrf2 and the silent information regulator 2 homolog 1 (SIRT1), which have antioxidant actions [101].

In sum, oxidative stress and neuroinflammation are reciprocally promoting each other, perpetuating the conditions for the development of the depressive pathology. Thus, targeting one or both is a promising strategy for the reduction of depressive disorder symptoms.

## 5. Plant-Derived Antioxidant Molecules in MDD Treatment

Diet is clearly associated with depression risk: an antioxidant diet reduces depression risk while a pro-oxidant diet increases it [102]. Therefore, the protective or therapeutic potential of putative diet components is currently being studied. In particular, plant-derived compounds are attracting attention because of their natural origin, in addition to their high therapeutic potential associated with their antioxidant and anti-inflammatory actions.

One of the many diet components with antidepressant effects is the regular tea (camelia cinesis). Many different tea compounds can reach these effects, acting on different depression-related alterations. For example, the anti-inflammatory and antioxidant effects of tea polyphenols might be crucial in the depression risk reduction properties of tea [103] (Figure 2). However, relevant limitations in the potential use of these compounds for MDD treatment are their low bioavailability, instability and low intestinal absorption [104]. Stability and bioavailability of tea polyphenols could be improved by their incorporation into nanocarriers. To date, however, the effectivity of this strategy has only been proved in in vitro conditions [105].

Similarly, turmeric curcumin has anti-inflammatory and antioxidant actions that may reduce anxiety and depressive symptoms in patients with depressive disorder and receiving standard care [106]. Accordingly, curcumin was also able to reduce depressive-like symptoms in a stress-induced animal model of depression—an effect mediated by the inhibition of the NLRP3 inflammasome and the regulation of kynurenine and quinolinic acid levels, which are products of tryptophan degradation with neuroprotective and neurotoxic effects, respectively [107]. Another study showed that curcumin reduced depressive-like symptoms as well as diminished stress-induced ROS levels in the same animal model by increasing the antioxidant promoting transcription factor Nrf2, which upregulates the expression of several antioxidant enzymes [108]. 

Another common culinary ingredient that has high curcumin content is saffron (*Crocus sativus*). It has been postulated that its use as a diet supplement may reduce oxidative stress, assessed by the reduction in MDA levels and the increase in the total antioxidant capacity of unhealthy individuals [109]. Saffron administration was more effective than placebo, and it was suggested to be as effective as antidepressant drugs in the treatment of depressive symptoms [110,111,112]. These antidepressant effects are believed to be attributable to saffron’s antioxidant and anti-inflammatory actions [110,111,112]. Other bioactive compounds in saffron that may exert antidepressant actions are crocins, crocetin, picrocrocin, and safranal. On the other hand, combining saffron with a low-dose curcumin treatment did not enhance treatment efficacy [113], suggesting that all these bioactive components act via similar mechanisms and reduce oxidative stress. Two recent metanalyses showed that curcumin effects are better than those of placebo, and it was concluded that curcumin supplementation may be beneficial for depressed patients as an additional intervention to standard treatment; however, larger randomized controlled trials are needed to improve the low-quality evidence that is currently available [106,114]. Interestingly, curcumin is more beneficial in the treatment of depressive symptoms in patients diagnosed with atypical depression [113], a depression subtype more frequently seen in women who have higher suicidal risk [115]. It is worth noting that atypical depression is associated with increased lipid peroxidation as compared to melancholic depression [116]. As in the case of polyphenols from tea, orally administered curcumin has low bioavailability due to their low intestinal absorption and biotransformation [117]. Its bioavailability can be enhanced by different delivery systems, including its combination with piperine to inhibit its biotransformation, or with lecithin to improve gastrointestinal absorption, among others [118]. The effectivity of the administration of curcumin in different delivery systems has yet to be tested. For now, its combination with piperine has not shown to be more effective than curcumin alone. However, the higher doses (1 g per day) did appear to be more effective than the lower doses [119].

Another intensively studied herbal treatment derived from Chinese medicine that is commonly used for depression treatment since ancient times is St. John’s wort (*Hypericum perforatum* L). It is reported to possess anti-inflammatory, antioxidant, antifatigue, and antidepressant capabilities [120,121]. Its main bioactive components are hyperforin, rutin, and melatonin. St John’s wort has been shown to be non-inferior to standard pharmacological treatment (SSRI) in its efficacy and safety for patients with mild-to-moderate depression in two studies [122,123]. However, longer randomized control trials are needed to establish the long-term effects of St John’s wort, as well as trials including individuals with severe depression in order to establish its efficacy in this kind of patients.

Turra Hypericum triquetrifolium is a plant closely related to Hypericum perforatum, which shares many biologically active compounds. It is reported to have potent antioxidant activity related to its methanolic extractable components and high hypericin content [124].

Accordingly, its administration to chronically stressed rats markedly increases hippocampal BDNF levels and reverts the stress-induced cognitive deficit [124]. As discussed for tea polyphenols and curcumin, the hypericum perforatum extract has a poor pharmacokinetic profile with low bioavailability and a reduced penetration of the blood–brain barrier. In spite of this, it induces its antidepressant effect in 4–6 weeks [125], a time period similar to standard antidepressant drugs. However, a major problem with all these kinds of medicinal plant extracts is that they are commercially available under different categories in different counties, which implicates different regulations [126], as, for example, with herbal supplements. Herbal supplements are not regulated by the FDA; therefore, they may be highly variable in their composition, greatly affecting their possible therapeutic effects [127]. Thus, the efficacy and safety of supplements offered in pharmacies over the counter is questionable.

Finally, ascorbic acid or vitamin C is a potent antioxidant synthesised by plants and most animals but not by primates, which is why for humans, this is an essential vitamin in the diet. Nevertheless, its consumption is decreasing worldwide, with a correlated increase in health problems associated with its deficiency. Levels of vitamin C are higher in the brain than in the periphery, and diseases associated with low ascorbic acid levels in the plasma are mainly related to the central nervous system [128]. In fact, there is accumulating evidence showing that ascorbic acid supplementation can reduce physiological alterations and symptomatology of neuropsychiatric disorders [129]. In stress-induced depressed-like rats, vitamin C levels are reduced in the brain, and lipid peroxidation is increased. However, the administration of a single dose of ascorbic acid can reverse depressed behaviour in these animals [130,131] in a way that is comparable to the ketamine effect observed in chronic cortisol-injected depressed-like mice [132]. Moreover, vitamin C intake through the diet is inversely correlated with depressive symptoms in middle aged women [133]. A recent metanalysis reported that vitamin C does not further reduce depressive symptoms in patients taking antidepressant drugs, but in individuals with subclinical depression, vitamin C supplementation is effective in inducing mood improvements [134]. The complex pharmacokinetic of vitamin C should be taken into consideration when studying the effects of vitamin C supplementation in MDD patients since, for example, several diseases can affect its turnover and reduce its plasma concentration [135].

There is an increasing amount of evidence showing the potential of plants and plant derivates in the treatment of neuropsychiatric disorders, including depression and specific depression symptoms such as anxiety or a depressed mood. This opens possibilities for alternative treatments that may help patients who do not respond to conventional treatment or experience complications, associated risks, or unpleasant effects with these treatments. Thus, plant antioxidant-based treatment may offer safer alternatives for the treatment of depression, or more effective alternatives for the treatment of atypical depression (Figure 2).

## 6. Antioxidant and Anti-Inflammatory Potential of Mesenchymal Stem Cells in the Treatment of MDD

Mesenchymal stem cells (MSCs) have potential as a future tool in the treatment of depression. They are multipotent stromal cells able to self-renew and to differentiate into various cell lineages mainly of mesodermal origin, favouring the regeneration of the damaged tissues [136]. MSCs can be isolated from different tissues of an adult organism, including bone marrow, adipose tissue, deciduous teeth, and menstrual blood [137]. MSCs have been tested as autologous and heterologous treatments for different pathologies and injuries, many of them associated with an increase in inflammation and oxidative stress [138]. For example, the systemic administration of MSCs produced immunomodulatory actions and reduced neuroinflammation in an animal model of stroke [139], but also in clinical trials [140]. In the same sense, it was recently reported that human MSCs—which had been activated in vitro by supplementing the culture medium with proinflammatory factors in order to increase their anti-inflammatory and antioxidant potential—when intracerebroventricularly administered to ethanol-drinking rats, they were able to dramatically reduce voluntary ethanol drinking and supress relapse with the concomitant abolition of ethanol-induced neuroinflammation and oxidative stress [141]. It is well-accepted that the main mechanism of action of MSCs is related to the paracrine secretion of several therapeutic molecules with anti-inflammatory and antioxidant activity, known as secretome [142]. However, compared with the administration of living cells, MSC-secretome has the advantage that it can be intranasally administrated to efficiently reach the brain. In fact, a result similar to the intracerebroventricular administration of activated MSCs was obtained with the intranasal administration of secretome derived from activated MSCs in rats that voluntarily consumed ethanol or nicotine, thus inhibiting their chronic self-administration of the drugs and fully abolishing neuroinflammation and oxidative stress in both models [143].

In the specific case of major depression, it has recently been reported that the administration of MSCs obtained from mouse fat tissue in chronically stressed mice with depressive-like symptoms is able to revert the depressive behavioural phenotype by remediating microglial activation and reducing the expression of inflammatory factors. Additionally, MSC administration also promoted the expression of BDNF, TrkB, and Nrf2 [144], thus increasing the antioxidant defence capacity.

Plasma Hydrogen sulphide (H2S) levels are reduced in depressed patients and are directly correlated with depression severity [145]. H2S was first known as a toxic gas, but it is now considered to be part of the same endogenous gas transmitter family as nitric oxide and carbon monoxide and is synthesized by mammalian tissues [146]. It is known to have potent anti-inflammatory and antioxidant capabilities [147]. MSCs produce H2S, and in turn, H2S is essential for the maintenance of MSC function, increasing their survival and proliferation in the context of inflammatory and oxidative conditions [148]. Furthermore, H2S increases the expression of Sirt1 and can revert the depressed-like symptoms induced by sleep deprivation in rats, reducing the levels of the pro-inflammatory cytokines IL-1β, IL-6, and TNF-α and the CCL2 chemokine, as well as increasing the levels of anti-inflammatory cytokines in the hippocampus [149].

MSCs have been proposed to reduce oxidative injury via several mechanisms, including: (i) scavenging free radicals, (ii) enhancing host antioxidant defences, (iii) modulating the inflammatory response, (iv) augmenting cellular respiration and mitochondrial functions, or (v) donating their mitochondria to protect damaged cells [150]. Most of these antioxidant actions can be replicated by the administration of the MSC-derived secretome [151], which contains soluble molecules but also small microvesicles called exosomes containing a broad set of bioactive molecules, including proteins, lipids, and nucleic acids. In this sense, it has recently been reported that MSC-derived exosomes can putatively reverse LPS-induced mitochondrial disfunction in astrocytes and reactive astrogliosis in mice by inhibiting the Nrf2-NF-κB signalling pathway [152].

Thus, MSCs produce a broad set of antioxidant and anti-inflammatory actions that can help to improve a dysregulated oxidative/antioxidant equilibrium commonly seen in MDD (Figure 2). These effects can most probably be efficiently achieved by the non-invasive administration of exosomes or secretomes derived from MSCs.

## 7. Conclusions

We have shown here that an antioxidant effect is a common property of several very different therapeutic approaches for the treatment of depression, and that oxidative stress is present in depressed patients and clearly related to their symptomatology.

The intake of natural compounds with antioxidant activity is promising as a strategy for avoiding or delaying the appearance of depressive symptoms or as a safer alternative treatment compared to currently available drugs. Nevertheless, more evidence is needed to endorse their antidepressant effects as well as stricter regulations in the production and characterization of these natural compounds. Likewise, more research is needed to test if the antioxidant effects of these natural compounds and antidepressant drugs are sufficient to support their antidepressant actions.

MSCs and their acellular derivatives (secretome or exosomes) have been proved to be effective in ameliorating depressive symptoms in animal models of depression, converting them into a promising future tool in depression treatment. The characterization of exosome contents and the antidepressant potential of their cargo is an important further step in the investigation that would lead to their future use as antidepressants. Finally, an understanding of the mechanisms that could regulate exosome cargo destination can help in the final goal of creating customized therapeutic exosomes for the treatment of depression in a more effective and safer way.

## Figures and Tables

**Figure 1 antioxidants-11-00540-f001:**
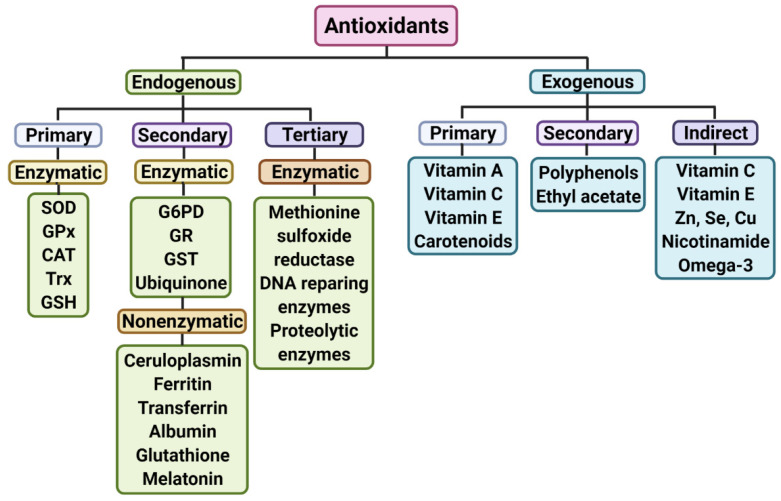
Endogenous and exogenous functional components of the antioxidant system in humans. The figure shows antioxidant molecules organized according to its origin that can be endogenous, which are synthetized by the organism, or exogenous, which have to be consumed in the diet. In addition to their source (endogenous or exogenous), antioxidants may be classified according to their antioxidant action into primary, secondary, and tertiary. Primary antioxidants are chain-breaking antioxidants that accept free radicals terminating the propagation of oxidative reactions and transform free radical species into more stable and less reactive products. Secondary antioxidants are radical scavenging molecules, and they have a preventive role in suppressing chain reaction initiation. Tertiary antioxidants are enzyme systems that can repair biomolecules that have been damaged by oxidation. Additionally, antioxidant action could be direct or indirect. Indirect antioxidants enhance many of the direct primary and secondary antioxidants. Finally, antioxidants could be enzymes or non-enzymatic molecules. SOD: superoxide dismutase, CAT: catalase, GPx: glutathione peroxidase, Trx: thioredoxin, GR: glutathione reductase, GSH: reduced glutathione, G6PD: glucose 6 phosphate dehydrogenase, GST: glutathione S transferase.

**Figure 2 antioxidants-11-00540-f002:**
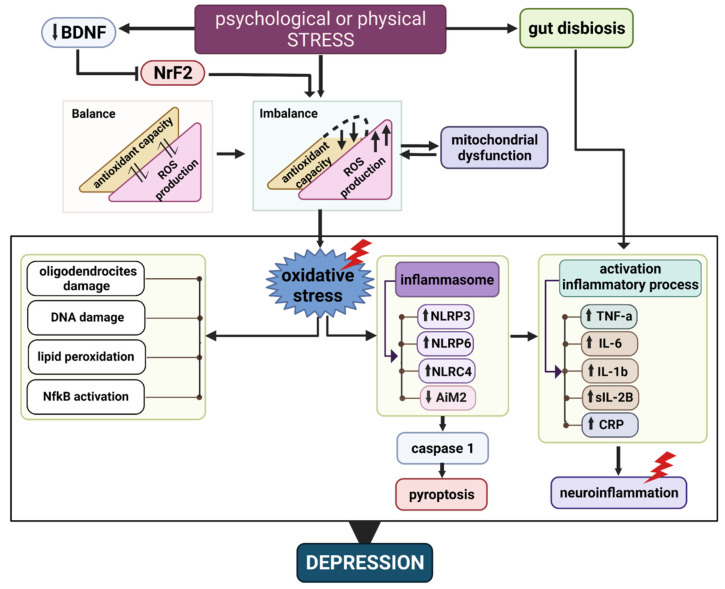
Interaction between oxidative stress and neuroinflammation at the onset of major depressive disorder. Figure shows that psychological and/or physical stressors can trigger the pathophysiology associated with major depression. Once the limit of the brain’s antioxidant capacity has been exceeded, oxidative stress prevails, inducing neuroinflammation and the deterioration of brain cells, which over time leads to the induction of the main phenotype associated with major depressive disorder. Red rays show the possible targets of plant-derived extracts and acellular products derived from mesenchymal stem cells.

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
