# Peer review of "Antioxidant Biomolecules and Their Potential for the Treatment of Difficult-to-Treat Depression and Conventional Treatment-Resistant Depression"

_antioxidants, 2022, doi:10.3390/antiox11030540_

Round 1

Reviewer 1 Report

This is an interesting review of the potential effect of antioxidants in treating major depressive disorders. Overall, this is an interesting paper that is well written. The comments are to enhance the paper:

  1. It would be helpful to categorize the types of antioxidants. A table summarizing major categories of antioxidants are helpful and whether they are endogenous sources or exogenous sources.
  2. Its unclear from the review what the pharmacodynamic response might be for the antioxidants reviewed. It would be useful to discuss them in context of timing, dose, degradation rates and types of MDD that are helpful or not helpful. Perhaps another table will be useful.

Author Response

Reviewer #1

Rev#1-Comment 1: This is an interesting review of the potential effect of antioxidants in treating major depressive disorders. Overall, this is an interesting paper that is well written.

Rev#1-Reply 1: Many thanks for your kind comment.

Rev#1-Modification 1: No modification needed.

Rev#1-Comment 2: It would be helpful to categorize the types of antioxidants. A table summarizing major categories of antioxidants are helpful and whether they are endogenous sources or exogenous sources.

Rev#1-Reply 2: Many thanks for this comment. We agree that a figure summarizing major categories of antioxidants could help the readers.

Rev#1-Modification 2:  A new figure (Figure 1) was incorporated. To put the figure more in context we have added a line referring to it in page 5 (lines 237 to 239).

Rev#1-Comment 3: Its unclear from the review what the pharmacodynamic response might be for the antioxidants reviewed. It would be useful to discuss them in context of timing, dose, degradation rates and types of MDD that are helpful or not helpful. Perhaps another table will be useful.

Rev#1-Reply 3: Many thanks for this interesting comment. Adding data of the pharmacodynamic response for the main antioxidants could help in understanding their real therapeutic potential.

Rev#1-Modification 3:  Pharmacodynamic data for the main antioxidants proposed were added in section 5, pages 9 and 10.

Reviewer 2 Report

In this manuscript, the authors have reviewed the possibility of antioxidant biomolecules serving as effective therapeutic interventions for people suffering from major depressive disorder (MDD). While depression related ailments are known to affect a huge percentage of the world population, Covid 19 related circumstances have further exacerbated the effect. Hence addressing the widely prevalent MDD is important for the general well-being of the world population. The prevalent medications to treat MDD can be quite aggressive in nature leading to several adverse side-effects for which many of these are discontinued by patients. The authors note that many of these antidepressants have a common mode of action, that is, by reducing oxidative stress. Yoga has been reported to diminish oxidative stress that also leads to increased BDNF levels. Similarly electric acupuncture leads to increase in glutathione levels, the latter being a very effective antioxidant. Moreover, diet control involving omega 3 fatty acids can also reduce oxidative stress induced neuroinflammation and also have an effect on the telomere length, the latter being associated with stress. Antioxidant capacity can be severely reduced in patients suffering from depression with the mitochondrial function also being affected. The review also brings out the potential that some plant derived natural compounds that have shown antioxidant activity, have in combating MDD. Finally, the antioxidant and anti-inflammatory effects of the mesenchymal stem cells (MSC) through their secretome as a non-conventional form of therapy has been highlighted

In summary, this review puts forth the idea that the common link amongst the different treatments for MDD is reduction of oxidative stress through the antioxidant effects of several molecules. This manuscript thus compiles important observations related to antioxidant effect-based treatment of MDD and is going to be an important contribution in the field.

The manuscript should be accepted after the following points have been addressed:

  1. Does ECT (Electro-convulsive therapy) have any effect on the oxidative stress?
  2. On page 3, the benefits if yoga have been mentioned. However, I am not sure how the effects of yoga can be tested in animals, as the authors have mentioned on page 3, last paragraph.
  3. It is known that protein aggregation can give rise to neurodegenerative diseases, the latter also involving ROS over-production. Are there data to show that patients having MDD, are more prone to develop neurodegenerative ailments or vice versa.

Author Response

Reviewer #2

Rev#2 Comment 1: In this manuscript, the authors have reviewed the possibility of antioxidant biomolecules serving as effective therapeutic interventions for people suffering from major depressive disorder (MDD). While depression related ailments are known to affect a huge percentage of the world population, Covid 19 related circumstances have further exacerbated the effect. Hence addressing the widely prevalent MDD is important for the general well-being of the world population. The prevalent medications to treat MDD can be quite aggressive in nature leading to several adverse side-effects for which many of these are discontinued by patients. The authors note that many of these antidepressants have a common mode of action, that is, by reducing oxidative stress. Yoga has been reported to diminish oxidative stress that also leads to increased BDNF levels. Similarly electric acupuncture leads to increase in glutathione levels, the latter being a very effective antioxidant. Moreover, diet control involving omega 3 fatty acids can also reduce oxidative stress induced neuroinflammation and also have an effect on the telomere length, the latter being associated with stress. Antioxidant capacity can be severely reduced in patients suffering from depression with the mitochondrial function also being affected. The review also brings out the potential that some plant derived natural compounds that have shown antioxidant activity, have in combating MDD. Finally, the antioxidant and anti-inflammatory effects of the mesenchymal stem cells (MSC) through their secretome as a non-conventional form of therapy has been highlighted.

In summary, this review puts forth the idea that the common link amongst the different treatments for MDD is reduction of oxidative stress through the antioxidant effects of several molecules. This manuscript thus compiles important observations related to antioxidant effect-based treatment of MDD and is going to be an important contribution in the field.

Rev#2 Reply 1: Many thanks for your kind description of our work.

Rev#2 Modifications1: No modification needed.

Rev#2 Comment 2: Does ECT (Electro-convulsive therapy) have any effect on the oxidative stress?

Rev#2 Reply 2: This is an interesting question. It has been reported that in bipolar depressed patients that respond to ECT treatment, the oxidative stress marker malondialdehyde (MDA) is reduced by the ECT treatment [Lv, Q et al. Int. J. Neuropsychopharmacol 2020, 23, 207], suggesting that antioxidative effect of ECT could be relevant for its antidepressant effect.

Rev#2 Modifications 2: This information was incorporated to the manuscript (page 5, lines 212-215).

Rev#2 Comment 3: On page 3, the benefits if yoga have been mentioned. However, I am not sure how the effects of yoga can be tested in animals, as the authors have mentioned on page 3, last paragraph.

Rev#2 Reply 3: We apologize for the mistake. We have rephrased that sentence to improve its legibility.

Rev#2 Modifications 3: In page 3, lines 142-144 we modified that sentence as follows “Acupuncture as opposed to psychological therapy, yoga or meditation can be tested in animal models, which are valuable tools for studying its possible mechanisms of actions.

Rev#2 Comment 4: It is known that protein aggregation can give rise to neurodegenerative diseases, the latter also involving ROS over-production. Are there data to show that patients having MDD, are more prone to develop neurodegenerative ailments or vice versa?

Rev#2 Reply 4: Many thanks for this interesting question.  Indeed, neurodegenerative diseases are bidirectionally related with depression (Galts, C et al, Neurosci. Biobehaw. Rev. 2019, 102, 56) and oxidative stress is part of their shared pathophysiology (Wadhwa, R et al, Curr. Pharm. Des. 2018, 24, 4711). Comorbid MDD in neurodegenerative diseases such as Parkinsons, Alzheimers and Hungtintons diseases respond poorly to standard antidepressant treatments and are in higher risk of side effects (Galts, C et al, Neurosci. Biobehaw. Rev. 2019, 102, 56). Therefore, the search for alternatives of treatment for depression in patients with neurodegenerative disease is urgent and oxidative stress appears as an interesting therapeutic target.

Rev#2 Modifications 4: This information was incorporated to the manuscript (page 5, lines 216-222).
